# Microbial Interactions in Rearing Systems for Marine Fish Larvae

**DOI:** 10.3390/microorganisms13030539

**Published:** 2025-02-27

**Authors:** Vasiliki Paralika, Pavlos Makridis

**Affiliations:** Department of Biology, University of Patras, 26504 Rio, Greece; vparalika980@gmail.com

**Keywords:** live food web, rotifers, Artemia, bacterial colonization, microalgae, biofilm, rearing systems

## Abstract

This article reviews the scientific literature discussing the microbial interactions between water microbiota, live food microbiota, fish larvae immune system and gut microbiota, and biofilm microbial communities in rearing systems for marine fish larvae. Fish gut microbiota is the first line of defense against opportunistic pathogens, and marine fish larvae are vulnerable to high mortalities during the first weeks after hatching. The bacterial colonization of fish larvae is a dynamic process influenced by environmental and host-related factors. The bacteria transferred to larvae from the eggs can influence the composition of the gut microbiota in the early stages of fish. Fish larvae ingest free-living microorganisms present in the water, as marine fish larvae drink water for osmoregulation. In marine aquaculture systems, the conventional feeding–rearing protocol consists of zooplankton (rotifers, Artemia, and copepods). These live food organisms are filter-feeders. Once transferred to a new environment, they quickly adopt the microflora of the surrounding water. So, the water microbiota is similar to the microbiota of the live food at the time of ingestion of live food by the larvae. In aquaculture rearing systems, bacterial biofilms may harbor opportunistic pathogenic bacteria and serve as a reservoir for those microbes, which may colonize the water column. The methods applied for the study of fish larvae microbiota were reviewed.

## 1. Introduction

During the last decades, one of the focus areas in aquaculture microbiology has been the study of the microbiota of marine fish larvae. The microbiota of fish is crucial for host health and development [1], but relatively little is known about the assembly of fish microbiota during the first weeks after hatching. Gut microbiota is an important obstacle for the colonization of opportunistic pathogens in the larval gut, while marine fish larvae are vulnerable to high mortalities during the first weeks after hatching caused by the proliferation of opportunistic bacteria. The optimization of the rearing conditions that promote the development of a stable, diverse, and resilient microbiota requires the determination of the factors influencing microbial colonization [2,3]. Bacteria colonize fish skin, fins, and gills, and several studies show that members of many bacterial taxa colonize fish surfaces [4]. The bacterial colonization of fish larvae is a dynamic process influenced by environmental and host-related factors. Detailed studies of the early fish microbiota have been conducted in a few species, with a focus on the gut microbiota [5], as it is crucial for nutrient absorption, digestion, and overall health. These species are gilthead seabream (*Sparus aurata*) [6,7], yellowtail kingfish (*Seriola lalandi*) [8], cod (*Gadus morhua*) [9], Atlantic salmon (*Salmo salar*) [10], Senegalese sole (*Solea senegalensis*) [11], greater amberjack (*Seriola dumerili*) [12,13], and giant grouper (*Epinephelus lanceolatus*) [14]. This article reviews scientific literature discussing the interactions between water microbiota, live food microbiota, fish larvae immune system and gut microbiota, and biofilm microbial communities in rearing systems for marine fish larvae (Figure 1).

Initially, bacteria in larvae may originate from the mother fish since the fish egg microbiota influences the larval microbiota [2,15] and may affect subsequent larval performance [16]. The transfer of bacteria from the mother to fish eggs is a process known as vertical transmission [17], and it occurs during the development of the eggs within the ovaries. These bacteria originating from the mother fish may include members of the phyla Proteobacteria, Firmicutes, Actinobacteria, and Bacteroidetes [18]. Glycoproteins present on the surface of the eggs may contribute to the development of species-specific bacteria during this stage [19]. Different fish species may have different dominant bacterial groups that are vertically transmitted. In yellowtail kingfish, through the larval developing stages from 0 days after hatching (dah) until 53 dah, the core larval microbiome included five Operational Taxonomic Units (OTUs): OTU205 (*Lactobacillus* sp.), OTU8677 (*Chryseobacterium* sp.), OTU327 (*Psychrobacter* sp.), OTU263 (*Acinetobacter* sp.), and OTU153 (*Corynebacterium* sp.) that were neither found in the water nor the first rotifer feed, suggesting that some bacteria present in the larvae might originate from the eggs and persist throughout development [20]. Hansen and Olafsen observed differences in the bacterial colonization of halibut (*Hippoglossus hippoglossus*) and cod (*Gadus morhua* L.) eggs, recognizing that interspecific variation existed [21]. The bacteria transferred to larvae from the eggs can influence the composition of the gut microbiota in the early life stages of fish, as upon hatching, sterile larvae take in the chorion-associated bacteria, which become the first colonizers of the developing gastrointestinal tract [22]. Moreover, fish eggs are eventually released into the water, allowing water-associated bacteria to colonize the egg surface [23]. Also, the first bacterial colonizers of the epidermal mucosal layer of larvae originate from bacteria present in the water and the eggs [2,15]. The gut is colonized by a microbial community before the fish start exogenous feeding [6,7,8,14,24,25,26] after the opening of the mouth, which happens hours/days after hatching, depending on the species and culture conditions [27].

Fish larvae drink seawater to osmoregulate, and in doing so, they ingest microorganisms present in the water, so water microbiota influence fish microbiota even before feeding starts. The first feeding food web for a fish-rearing tank is a complex ecosystem [28]. Gill and skin microbial communities increase in diversity with fish age, and they obtain a similar microbiome to water and tank biofilm microbiota [29]. At the same time, fish microbiota may influence water and microbiota and vice versa [30]. In the rearing of marine fish larvae, live feed organisms including rotifers, Artemia, or copepods are used. Gut microbial communities of fish larvae are affected by the type and the bacterial composition of live feed [22,31]. Zooplanktonic prey organisms, microalgae, and other larvae interact continually with each other and influence the microbiota of the larvae (Figure 1).

## 2. Roles of Fish Larvae Microbiota

Fish larvae microbiota plays a crucial role in the development and health of marine fish larvae. One important aspect is the contribution of microbiota to nutrient absorption and digestion [32,33,34]. Bacterial colonization improved the nutritional state of cod larvae, evident at both transcriptional and micromorphological levels and in terms of fish growth rate [32]. Members of fish larvae microbiota are capable of fermenting complex carbohydrates in the larval diet, producing short-chain fatty acids (SCFAs) [35]. SCFAs can serve as an additional energy source for the larvae, contributing to their overall nutritional status and reducing the intestinal pH, making the environment unfavorable for some potential pathogens, and protecting the fish from bacterial infections [36]. Marine fish larvae often consume prey with exoskeletons containing chitin (such as *Artemia*, copepods, and rotifers). Chitin is a complex polysaccharide, and gut microbes may assist in the digestion of chitin [37]. Dietary chitin modifies fish gut microbiota due to its prebiotic, antibacterial, and immunomodulatory properties and increases gut bacterial richness and the amount of beneficial chitin-degrading bacteria [38].

Members of the larvae microbiota may synthesize vitamins that are essential for the growth and development of the larvae [39]. Indigenous gut microbes, able to synthesize vitamin B12, exhibited a probiotic effect on host resistance to pathogen infections [40]. The presence of a well-established microbiota may influence the development of the gastrointestinal tract in fish larvae [41], including the morphology of the gut. Optimal gut development facilitates efficient nutrient absorption [1].

Fish possess elements of both innate and adaptive immunity. Fish larvae do not have the ability to develop specific immunity. For specific immunity, they rely on immunoglobulins from the mother. Innate defense mechanisms include the presence of physical barriers, humoral factors, cellular defenses, and inflammatory processes. In cod larvae, gene expression analysis indicated that certain elements of the innate immune system were activated in the larval stage of the fish but were subsequently regulated by host–microbiota interactions [32]. As larvae grow, microbiome stability could be under the control of the innate immune system. An effectively modulated immune system can be established by a balanced and diverse microbiota [42]. Teleost immune responses towards bacterial infections initially follow a general pattern of responses, including pathogen recognition, signal transduction, the initiation of inflammatory reactions, the production and release of innate effector molecules, and the subsequent establishment of adaptive responses with the involvement of T and B cells [43]. Healthy microbiota allows immune system maturation and the modulation of inflammation, both of which contribute to immune homeostasis. Thus, immune equilibrium is closely linked to microbiota stability and to the stages of microbial community succession during the host development.

The presence of diverse microorganisms in the gut microbiota provides a source of antigens that can be recognized by the developing immune system of fish larvae. Microbial antigens contribute to the training and maturation of immune cells, including antigen-presenting cells (APCs), which play a role in immune response initiation. Cytokines that regulate innate immunity are produced primarily by macrophages in response to microbial antigens. A balanced and diverse microbiota contributes to the maintenance of gut homeostasis, preventing inflammation and supporting the development of a tolerant immune system [44]. Also, fish mucosal barriers draw many benefits from their symbiotic microbial communities, and therefore, the mucosal immune system of fish plays a key role in maintaining this symbiosis [45]. The innate immune system, which fish larvae depend on before the maturation of the adaptive immune system, employs receptors known as pattern-recognition receptors to identify microbial-associated molecular patterns generated by members of the bacterial microbiome. Toll-like receptors (TLRs), extensively researched pattern-recognition receptors, can identify microbial components such as lipopolysaccharides, peptidoglycans, nucleic acids, or flagella that, when activated, encompass the production of pro-inflammatory cytokine and chemokine genes [5].

Fish larvae microbiota offer protection against pathogens through various mechanisms that help maintain a balanced microbial community and support the overall health and viability of the larvae in rearing conditions. A first line of defense against pathogens is provided by bacteria on larval skin and mucosal surfaces of their intestine. Specifically, these bacteria compete for attachment sites and available nutrients with incoming pathogens, reducing the colonization and proliferation of these pathogens and thereby their ability to cause disease [46,47]. Some microbes produce antimicrobial substances, such as bacteriocins, which are compounds that are synthesized by microorganisms to kill closely related species [48], or organic acids produced either through the microbial fermentation of carbohydrates from various bacterial species in different metabolic pathways and conditions [49] or as secondary metabolites that inhibit the growth or kill unwanted marine pathogens such as tropodithietic acid (TDA), a broad-spectrum potent antibiotic produced by marine bacteria of the Roseobacter clade [50,51].

Gut microbial diversity has been utilized as a biomarker of fish health and metabolic capability [52]. Low diversity and the instability of microbiota are closely related to fish disease, as the chances for an invader to establish a niche in a high diverse environment are significantly diminished, and it is more likely that in these environments, there are members from the community with antagonistic activity [16]. Moreover, numerous disruptions, including environmental stress, can cause illness in fish by upsetting the gut microbiota and increasing the pathogen’s virulence [52].

Many of the bacteria colonizing the larvae are commensal or beneficial for the fish; however, interactions with pathogens or opportunistic bacteria can result in disease or death [15]. Bacterial taxa, such as *Vibrionales*, *Actinomycetales*, *Alteromonadales*, and *Flavobacteriales*, include opportunistic, rapidly growing bacteria, which dominated the water and larval microbiota in reared cod larvae [32]. This could be the outcome of the water’s r-selection [28], as the microbe–microbe competition in the water is decreased when incoming water with a lower carrying capacity is added to the rearing system, which promotes the growth of opportunistic, rapidly growing species (r-strategists). The genus *Vibrio*, belonging to the family *Vibrionaceae* and the class *Gammaproteobacteria*, comprises Gram-negative, chemoorganotrophic facultative anaerobic bacteria. *Vibrio* species exhibit high growth rates and thrive in warm (>18 °C) marine and brackish waters and alkaline conditions (pH 6.5–9.0). Some *Vibrio* species cause disease in humans or fish, but the majority are non-pathogenic species [53]. High numbers of *Vibrios* in seawater, live prey, or tank surfaces represent a potential hazard for cultured organisms. *V. anguillarum* is usually found in live prey organisms [54]. *V. anguillarum* does not colonize the gut of turbot larvae but may infect the epidermis of the larvae [55,56]. *V. anguillarum* could colonize plastic, sintered glass, and ceramics [57]. *Aeromonas veronii* is an opportunistic fish pathogen that caused disease in farmed European seabass in Greece during the past few years [58]. *V. harveyi* has been isolated from all rearing stages during periods with rising temperatures, common in summer and autumn, and it has been co-isolated with *Aeromonas veronii* [59] from farmed sea bass and sea bream in Greek aquaculture sites. *V. alginolyticus* caused disease in farmed sea bass at low temperatures (15–17 °C) in Greece [60]. Bacteria colonize the gut when fish eat live feeds or drink seawater [61]. Pathogenic *Vibrio* spp. form biofilms on different biotic and abiotic surfaces [62].

Control of bacterial infections in larval rearing systems may involve the management of potentially harmful bacteria, such as *Vibrio* spp., in the seawater of the rearing system. Seawater serves as a common medium for larvae, live feed, microalgae, and the rearing environment (including sediment, tank walls, pipes, etc.), where its microbial dynamics interact with those of the larvae, live feed, and environmental surfaces.

## 3. Factors Influencing Fish Larvae Microbiota

### 3.1. Host Selection and Developmental Stage

At the onset of first feeding, there is an increase in bacterial diversity and a change in community species composition. The bacterial dynamics associated with the early developmental stages of fish larvae are complex and influenced by various factors, including the surrounding environment, maternal influences, diet, and developmental changes in the larvae. As fish larvae develop, their digestive and immune system changes, which can influence the dynamics of the gut-associated bacterial communities. During development, gut morphology and function change influencing bacterial colonization patterns, including the types and the abundances of bacteria. The bacterial diversity in fish larvae tends to increase gradually with age until it peaks when approaching the juvenile stage [5]. In gilthead seabream, the lowest bacterial richness was found at 1–5 days after hatching (dah) [6,7], whereas, in cod larvae, the lowest bacterial richness was observed at 17 dah [9], while in greater amberjack, the lowest bacterial richness was shown at 9 dah, with a significant effect of age. It gradually increased to 31 dah [12]. In a study with yellowtail kingfish, the bacterial load of developing larvae significantly increased up to 14–18 days after hatching, then remained stable. High bacterial richness and diversity in *S. lalandi* larvae were observed as early as one day after hatching, while the larvae were still using their yolk-sac reserves [8]. The life cycle stage (from parr to adult stage) in wild Atlantic salmon had a large impact on the composition of intestinal bacterial communities, rather than location, driven, as was hypothesized, by deterministic and stochastic factors [63]. Host development was the driving force behind shifts in diversity, community composition, indicator taxa, and potential function in *Seriola dumerili* larvae microbiome succession [13].

Extrinsic (diet, water quality, rearing environmental physicochemical parameters, etc.) and intrinsic (trophic level, genetic background, gender, age, etc.) factors modulate fish larval microbiota. During the early larval developmental stages, the gut microbial community is generated by species-specific selection [64]. The presence of a core microbiota in fish species, reared in different environments [65] or across different dietary conditions and time series [64], indicated that host selection is a major factor affecting microbial composition. In cod larvae, their microbiota was very different from microbiota in live feed and rearing environments, and this demonstrated a host selection [9,66]. Yan et al. [67] developed the theory of “fish gut island”, where host factors, and not the environment, act in a deterministic way leading to the establishment of the microbial community assembly of microbiome. In sturgeon, deterministic processes were stronger than neutral during the early gut microbiota assembly [68].

According to Vadstein et al. [28], the fish microbiome is assembled through the predictable deterministic and random stochastic process. Deterministic processes include fish–bacteria interactions (environmental filtering) and bacteria–bacteria interactions (competition and mutualism). Stochastic processes are driven by ecological drift and the dispersal of bacteria from the environment to the fish.

### 3.2. Diet

Fish larvae are carnivorous, and they eat various zooplanktonic organisms in the wild, primarily copepods [69]. In marine aquaculture systems, where species such as cod, halibut, turbot, seabass, gilthead seabream, yellowtail kingfish, greater amberjack, and grouper are reared, the conventional feeding–rearing protocol consists of zooplankton (rotifers, Artemia, and copepods) [70]. The production of highly nutritious live food is one of the most critical parts of successful larvae rearing, as it influences not only larvae growth but also the development of deformities, malpigmentation, as well as overall health and hence survival [70,71]. Fish larvae can only consume small prey items, but as they mature, their larger mouth gaps and improved swimming abilities enable them to catch and consume a greater variety of prey [72]. Like mammals, fish gut microbiota is categorized as either autochthonous (native) or allochthonous (from other sources) [73]. While allochthonous bacteria are transitory, associated with food particles, or proliferate in the gut lumen, autochthonous bacteria can colonize the host’s gut epithelial surface [31]. Given that the GI tract is one of the main entry points for various pathogens, it is crucial to assess the impact of dietary microbial communities on the intestinal microbiota of fish [74].

Since live food production in larval hatcheries has become more intensive, the spread of disease has become a serious concern. It is widely known that, in larval cultures, live feeds may transmit opportunistic and putative pathogenic bacteria [6,32]. A survey of microbiomes of farmed European seabass and gilthead seabream similarly reported high relative abundances of Vibrio in fish larvae and their live food (>20% relative abundance) and postulated that the Vibrios were transferred from live food to larvae [18], although there were no supporting microbial source tracking data. Numerous investigations have demonstrated a relationship between the bacterial flora in live food and the larvae [6,7,9].

Rotifers are used as live food for many marine fish species because they are easily mass-cultured, their body size is perfect as the first feeding item for a large number of species, and it is easy to enrich them with HUFAs [75]. Rotifers’ microbiota composition has been shown to be influenced by their nutrition [76,77], while different diets significantly altered the bacterial communities associated with rotifers [78]. Rotifer diets can affect colonization by Vibrio members in the guts of post-larval turbot [79].

Culture-independent studies of rotifers, based on PCR/DGGE methodology, could identify bacteria belonging to genera Marinomonas and Pseudoalteromonas [80] or family Rhodobacteraceae and genera Vibrio, Roseobacter, Alteromonas, and Acrobacter [81]. Host-associated microbiota in rotifers comprised mainly α-Proteobacteria (55.5% of the relative abundance), Rhodobacteraceae (31.7% relative abundance), and Phyllobacteriaceae (20.2% relative abundance) families. The γ-Proteobacteria class showed 20.9% relative abundance, consisting mostly of species affiliated with the Vibrionaceae (6.7% relative abundance), Enterobacteriaceae (3.7% relative abundance), and Oceanospirillaceae (2.9% relative abundance) families [7], with some OTUs being shared between larvae and rotifers. Since live food production in larval hatcheries has become more intensive, according to Verdonck et al. [82], the genus Vibrio is prominent in rotifers, and in the culture of some fish species, it has been documented that numerous Vibrio species produce large mortalities [83]. In the early life stages of red seabream (*Pagrus major*), α-proteobacteria, Flavobacteria, and γ-proteobacteria (excluding Vibrionaceae) were predominant in fish guts and in rotifers [84]. Similar trends have been noted in the early stage of yellowtail kingfish [8] cod [9], Atlantic halibut [85], Nile tilapia [86], and Senegalese sole [11].

Artemia, commonly known as brine shrimp, are branchiopod crustaceans, which are used worldwide as live food for marine finfish larval culture [75]. They make cysts that can be easily stored, and A. salina and A. franciscana are the most studied species [87]. Cysts can be decapsulated or disinfected before incubation to reduce initial bacterial burdens. This is crucial since bacterial loads can increase significantly during hatching and possibly again during enrichment [85,87]. The microbiota of feeding Artemia appears to be similar to the microbiota of rotifers and is dominated by Vibrio spp., Pseudomonas spp., and Cytophaga/Flavobacterium spp., although newly hatched Artemia nauplii have been found to be predominantly colonized by uncultured members of γ-Proteobacteria and Planctomycetales [82,88]. Microbiota associated with Artemia sp. nauplii consisted almost entirely of γ-Proteobacteria and more specifically of members of the Alteromonadaceae family (85.9% relative abundance), while Artemia sp. metanauplii showed dominance by species belonging to the α-Proteobacteria family Rhodobacteraceae (70.1% relative abundance) [7], with some OTUs being shared between larvae and Artemia. Vibrio species were the dominant bacteria associated with rotifers and Artemia according to early, cultivation-dependent studies [82]. The bacterial load in cultured nauplii of Acartia tonsa was estimated to be 5 × 10^2^ colony-forming units (CFU) per individual, which is lower than the bacterial load in Artemia and rotifers [85,89]. Nevertheless, high diversity in the culture and enrichment techniques results in increased differences in the numbers of culturable bacteria per rotifer or Artemia [90]. Bacteria in copepods are often found attached to the exoskeleton in large clusters, in the lumen of the gut, and in skeletal muscles [91]. The two main bacterial groups found in marine copepods are α-proteobacteria and γ-proteobacteria, while Actinobacteria, Firmicutes, Verrucomicrobia, Betaproteobacteria, and Bacteroidetes are found [70]. Vibrios were comparatively uncommon in high-throughput sequencing analysis while often being observed in studies that relied on culture-dependent methods [92]. Copepods are not passive carriers and selectively interact with Vibrios [93]. Copepods stimulate the growth of Rhodobacteraceae, Vibrionaceae, and Oceanospirillales in the surrounding water [94]. Furthermore, copepods host on their surfaces or within their bodies, members of Flavobacteriaceae and Pseudoalteromonadaceae. As a result, the copepod microbiome possesses the capacity to regulate the composition of the bacterial communities in seawater or associated with them.

### 3.3. Water Microbiota

Water quality includes physical and chemical characteristics such as temperature, dissolved gases, pH, oxygen levels, and mineral content. Together with live food microbiota and microalgae, water microbiota shapes the microbial profile of the reared larvae [95,96]. The water exchange rate may be restricted during the first days after hatching due to the fragility of the larvae.

Pathogenic bacteria may enter the rearing system with the inlet water, usually treated with UV irradiation or ozonation [97]. Recent studies focus on microbial water quality in rearing environments [98,99]. Microbiologically, water quality is not easy to define, as it is characterized by both the quantity and the quality of the microbial communities present. The quantitative part has traditionally been in focus, and the continuous reduction of the bacterial load is the main approach in marine fish hatcheries [35,100]. The qualitative characteristics of the water microbiota are, however, more important [28], which are nevertheless difficult to assess, except for the presence or absence of well-known pathogens. Mortalities during the rearing of marine larvae are frequently unrelated to known obligatory pathogens [16]. A theory that may be used to define microbial water quality is based on the r/K strategy; bacteria are assigned into two functional groups, r- and K- selected bacteria, respectively [28]. K-strategists, according to Andrews and Harris [101], grow slower, are in dense areas with increased competition for resources, and tend to create more stable communities. Microorganisms following the r- strategy are found under the opposite spectrum of these conditions, and opportunistic pathogens often follow this mode. This approach has been applied to assess and steer microbial environments in larval rearing, and it appears that opportunistic species are mostly responsible for parasitic host–microbe interactions [102]. It is therefore advantageous to establish a stable microbial community in the rearing system that is dominated by K-strategists, while opportunistic strategists will benefit from the use of antimicrobial agents and disinfection [103], which will create new niches. Therefore, criteria for microbial water quality may be developed using traits based on r- and K-selection [104]. One of the goals of recirculating the water is to use microbial populations to condition the water since it is important to maintain a beneficial microbial rearing environment from hatching and throughout the larval rearing period. When comparing the water communities of flow-through system (FT) and recirculating aquaculture system (RAS) tanks, the richness and phylogenetic diversity were higher in RAS [29].

The microbes in the rearing water influence fish larval microbiota more heavily than the microbes introduced through the feed [9,86,105]. Recent studies concern the impacts of the rearing water system, and specifically the participation of the water bacterial communities, on the effectiveness of larval microbial, colonization, and establishment [18,29]. Marine fish drink seawater to osmoregulate, which in turn provides a large source of potential microbes for gut colonization [19]. For gill and skin communities, the water column microbiota contributed the most for reared fish yellowtail kingfish (*Seriola lalandi*) and the gut “allochthonous” microbiomes of the fish digesta originate primarily from water sources rather than feed sources [29]. In the case of live food organisms used in larval rearing, these are zooplanktonic organisms, which are filter-feeders. Once transferred to a new environment, they quickly (20–60 min) adopt the microflora of the surrounding water. So, the water microbiota is similar to the microbiota of the live food at the time of ingestion of live food by the larvae. This is, in our opinion, the pathway by which water microbiota influences the microbiota of the gut of the larvae. Live food organisms filter bacteria from the water, and these are transported to the larvae through the live food. Some of these bacteria will be established in the larval gut. The number of bacteria that enter the gut is much higher than the number of bacteria that enter the gut through water-drinking activity.

### 3.4. Biofilm

After a solid surface is first exposed to seawater, dissolved organic matter adheres to it and creates a thin film (<100 nm) known as “molecular fouling” in a matter of seconds [106]. Biofilm formation typically begins with early colonizing bacterial groups, as these groups secrete a matrix of polymeric substances, where dynamic interactions like quorum sensing, cooperation, and competition take place in the early stages of formation [107]. Biofilms are communities of microorganisms that adhere to solid surfaces in the rearing system and are embedded in an organic matrix that they produce. The transportation and adhesion of free-living bacteria to a surface, cell growth, the creation of microbial aggregates of a few dozen to a few thousand cells, and the distribution of offspring cells into the water column are all essential processes in the formation of biofilms [108] serving as bacterial stocks as well [109].

Multiple factors influence the composition and structure of biofilm microbial communities, such as niche differences, water quality changes, the availability of inorganic nutrients and organic matter resources for organisms, water depth and biofilm age, substrate surface properties, and temperature situation [110]. The colonization of solid surfaces and the formation of biofilms is advantageous for some bacteria, as biofilms resist desiccation, improve antibiotic resistance, have a higher nutrient concentration, and show a better defense against predators [111]. In aquaculture rearing systems, bacterial biofilms may harbor opportunistic pathogenic bacteria and serve as a reservoir for those microbes that may colonize the water column, which could make them a constant source of bacterial infection for the rearing species [112]. Furthermore, biofilms can withstand the action of disinfectants [113], which makes it difficult to remove them from surfaces. Bacteria within the biofilm, on the other hand, are useful for recycling nutrients and eliminating unwanted ones from the system, reducing toxic inorganic nitrogen through nitrification and heterotrophic bacteria, and enhancing fish and shrimp growth and survival [114].

The main determinants of the species composition of the biofilm are substrate types, water quality, and grazing pressure [113]. Proteobacteria and Bacteroidetes are classically the dominant taxa in the tank wall biofilm, water, and biofilter reservoirs of RAS systems [112]. The addition of specific substrates with diatom-bacteria biofilms in rearing systems for larvae of *S. lalandi* showed that they are consumed by the larvae without negative effects, while positive effects on the viability of larvae were determined [115]. Additionally, the Rhodobacteraceae family accounted for most of the bacterial components found in the biofilm, with a relative abundance of 73.7%. In the biofilm microbiota in a sea bass hatchery, it was shown that microbial communities varied significantly between feeding phases, due to the introduction of live feed-microalgae. Members of Methylobacterium and family Rhodobacteraceae dominated, specifically members of the genus Phaeobacter [78], which are known to exhibit inhibitory activity against fish pathogens, such as Vibrio, producing antibiotic tropodithietic acid (TDA) [56,116], attaching to inert surfaces, and subsequently forming a biofilm composed of rosette-shaped microcolonies [117]. Culturable bacteria of this study included putative fish pathogenic species such as Vibrio spp. and Tenacibaculum spp. In a Mediterranean RAS system, where adult sea bass and sea bream were reared for 30 days, studying the composition and dynamics of the heterotrophic assemblages in the water column and biofilm, it was shown that biofilm communities stabilized in terms of abundance after 6 days, whereas their composition changed throughout the experimental period and only showed signs of stabilization after sampling at day 18, indicating that the maturation and stability for biofilm microbial community is a relatively long process [118]. The fish pathogen *Tenacibaculum discolor* colonized the tank biofilm and could eventually colonize the water column or the fish. Thus, understanding the establishment of rearing tank biofilm microbiota may contribute to larval health.

## 4. Methods Used to Analyze Fish Larvae Microbiota

The study of microbial communities in fish can be based on culture-dependent or culture-independent methods. Only 1% of the total microorganisms present in coho salmon (*Oncorhynchus kisutch*) and Atlantic salmon could grow on tryptic soy agar (TSA) [17,119]. Likewise, less than 1% of bacterial species in an environmental sample could be cultivated [120]. Thus, the use of culture-dependent methods leads to underestimating microbial diversity, and at the same time, it is a time-consuming, labor-intensive process (Table 1).

The use of the conventional culture-dependent method is related to the microbial identification of fish larvae samples, which involves the homogenization of the animal, spreading of the homogenate on a general purpose or selective culture agar in the laboratory, incubation, and quantification by counting the grown colonies and isolation of pure cultures of viable microorganisms followed by testing for multiple physiological and biochemical traits [121]. Fish-associated microbiota studies before 2000 were based on this approach [122]. In recent years, most studies have been based, however, on 16S rRNA gene sequencing to identify microbiota in fish, water, or biofilms [123]. The cultivation approaches are restricted to a specific group of bacteria that are easy to grow and enable the study of microbial physiology, metabolism, and interactions under controlled laboratory conditions but typically only uncover a narrow spectrum of microbial diversity [124] leading to the underestimation of microbial diversity. Another limitation is the inability to culture all microorganisms present in a sample, particularly those with specific requirements [125] or in a viable but non-culturable state [126]. Furthermore, several bacterial phyla have no cultivable members and are recognized only by the detection of their DNA by molecular methods [125]. Thus, the culture-dependent methods allow bacterial identifications to the species level and one of its main implications is that culture-dependent methods can validate findings from culture-independent approaches by confirming the presence of specific microorganisms and providing isolates for further study such as the isolation and identification of bacteria, which have been characterized as potentially probiotics [20,127].

Studies on fish larvae microbiota focus on explaining the fundamental processes for functioning, aiming to develop innovative approaches for managing microbial populations. This requires the identification and characterization of the microbial communities of all the different parts of the rearing systems. The massive quantities of microbes in these systems could only be detected by new molecular tools, which are referred to as culture-independent methods and involve directly extracting DNA or RNA from a sample without the need for microbial cultivation (Table 1). Next-generation sequencing (NGS) was introduced in the early 21st century [123]. NGS has made tremendous development by allowing the rapid and cost-effective creation of massive volumes of data, with superior accuracy in DNA sequencing compared to earlier Sanger methods. At the moment, all NGS approaches require library preparation. This protocol occurs after DNA fragmentation, where adapters are attached to the ends of each fragment. It is usually followed by a step of DNA amplification to result in a library that can then be sequenced by the NGS platform (e.g., MiSeq/Illumina, 454/Roche pyrosequencing, etc.) [128]. They are widely used for profiling microbial communities in environmental samples, including fish larvae, and studying microbial diversity, community structure, and functional potential. Additionally, they provide a comprehensive view of microbial diversity, including uncultivable and rare microorganisms. Also, they enable the high-throughput analysis of microbial communities, allowing for the simultaneous profiling of multiple samples [128]. Some limitations have also been demonstrated when NGS techniques are used to detect bacterial profiles, such as the inability to distinguish between live and dead microorganisms, leading to potential overestimation of microbial diversity [129], limited ability to study microbial physiology and interactions in the absence of cultured isolates [130], and reliance on DNA sequencing technologies, which may be costly and require bioinformatics expertise for data analysis [131,132].

## 5. Conclusions and Future Perspectives

The study of microbial interactions in larval rearing systems has focused on the factors that influence the microbiota of the larvae, possible approaches to increase microbial balance in larval rearing systems, and the use of probiotics in larval rearing to improve survival and growth of the larvae. In earlier days, these approaches and their effect on the microbiota of water, fish, and live food were monitored by classical microbiology, which was a very laborious and highly inefficient process. Now, with the use of molecular techniques, it will be easier to obtain some more clear answers to the same questions. In addition, with the emergence of new live food organisms, such as cirriped larvae or easily available copepods, the effect on fish larvae microbiota will be far more reliably assessed.

## Figures and Tables

**Figure 1 microorganisms-13-00539-f001:**
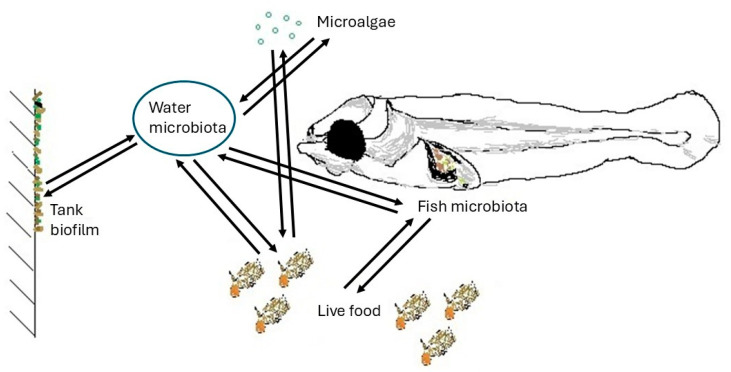
Microbial interactions in a rearing system for marine fish larvae. Live food organisms are filter-feeders and are thereby influenced by water microbiota and vice versa. Water microbiota is influenced by fish microbiota, microalgae added to the water, and tank biofilm, and vice versa. Fish larvae drink seawater and are thereby influenced by water microbiota and live food ingested.

**Table 1 microorganisms-13-00539-t001:** Methods for the study of fish microbiomes and their characteristics.

Method	Characteristics of the Method
Culture-based Methods (Classical microbiology)	Many bacteria are non-culturable. Identification based on cell morphology or physiological tests time- and labor-consuming and many times impossible to identify isolates at species level.
Biochemical Tests	Bacteria may have variable or weak reactions, leading to misidentification. Difficult to distinguish closely related species with similar metabolic profiles.
PCR-Based Identification (16S rRNA Sequencing, Multiplex PCR)	16S rRNA sequences may not differentiate closely related species. PCR inhibitors from environmental samples can affect results. Requires specific primers, which might not cover all bacterial taxa.
Metagenomics and Shotgun Sequencing	High cost and requires bioinformatics expertise. Short-read sequencing can lead to assembly errors. Contamination can mislead taxonomic classification.
Flow Cytometry and Fluorescence In Situ Hybridization (FISH)	Requires specific probes for known bacteria. Less effective for detecting rare or unknown taxa. Labor-intensive and requires expensive equipment.

## Data Availability

No new data were created or analyzed in this study. Data sharing is not applicable to this article.

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
