# Peer review of "Microbial Interactions in Rearing Systems for Marine Fish Larvae"

_microorganisms, 2025, doi:10.3390/microorganisms13030539_

Round 1
Reviewer 1 Report
Comments and Suggestions for Authors
The manuscript entitled Microbial interactions in rearing system for marine fish larvae has been revised and can be accepted with some changes. Kindly incorporate the changes before acceptance.
- Please recheck the intext citations carefully.
- What is the significance of the study should be highlighted in one line in abstract
- First Paragraph in introduction should provide the problem statement for the study. For instance, why the study is being conducted????
- The last paragraph is consisting of fragmented piece of work, it needs a coherent story to highlight the major hypothesis and objective of the study
In abstract: Fish microbiota is the first line of defense mechanism???? Is it sure? I think the mucus present on fish skin in the first line of defense.
This would be better if a research or problem question is provided in the introduction section, which highlight why the review study is being conducted?
It is better to provide a supplementary supported data for the highlighted sentences. For instance, lin 82, important aspect of microbiota is to absorb and digest nutrient. A study should be added, a study reported that phylum Firmicutes, plays a role in fat absorption in epithelial tissues of the intestine 10.1016/j.envres.2023.117002
I did not find any tabulated or illustrated data? It is better to provide some mechanistic illustrations.
Better to provide some limitations of the techniques or methods used for the identification of bacteria taxa. For instance, culture-independent methods have problems with sensitivity and cost effective.
What about DNA barcoding? Shotgun methods? I suggest a table should be made to present all the methods.
Future perspective is missing.
A summary of conclusion is necessary
My suggestion is to revise the manuscript, however, it must have some tables and mechanistic figures, a review study is incomplete without figures.
Author Response
- Please recheck the in-text citations carefully.
Answer. Thank you for pointing this out. We checked all citations carefully and found no errors.
- What is the significance of the study should be highlighted in one line in abstract.
Answer. Thank you for your comment. We inserted a line at the beginning of the abstract outlining the significance of this study.
- First Paragraph in introduction should provide the problem statement for the study. For instance, why the study is being conducted????
Answer. Thank you for your comment. We inserted a similar sentence as in the abstract at the end of the first paragraph of the introduction.
- The last paragraph is consisting of fragmented piece of work, it needs a coherent story to highlight the major hypothesis and objective of the study
Answer. Thank you for pointing this out. This paragraph was reconstructed to make it more coherent.
In abstract: Fish microbiota is the first line of defense mechanism???? Is it sure? I think the mucus present on fish skin in the first line of defense.
Answer. Thank you for your comment. This sentence was changed. When considering fish microbiota, we focus in general more to the microbiota of the gut.
This would be better if a research or problem question is provided in the introduction section, which highlight why the review study is being conducted?
Answer. Thank you for your comment. This question was asked by the reviewer previously and was answered.
It is better to provide a supplementary supported data for the highlighted sentences. For instance, lin 82, important aspect of microbiota is to absorb and digest nutrient. A study should be added, a study reported that phylum Firmicutes, plays a role in fat absorption in epithelial tissues of the intestine 10.1016/j.envres.2023.117002
Answer. We inserted the specific publication in the manuscript.
I did not find any tabulated or illustrated data? It is better to provide some mechanistic illustrations.
Answer. We have added a figure and table to the article.
Better to provide some limitations of the techniques or methods used for the identification of bacteria taxa. For instance, culture-independent methods have problems with sensitivity and cost effective.
Answer. We have added Table 1, where there is comparison of the different techniques in tabulated form.
What about DNA barcoding? Shotgun methods? I suggest a table should be made to present all the methods.
Answer. To our knowledge, DNA barcoding is not typically used for bacterial identification in fish larvae microbiota studies. Instead, amplicon sequencing (16S rRNA gene sequencing) and shotgun metagenomic sequencing are the preferred and appropriate methods for studying bacterial communities.
DNA barcoding involves sequencing a standardized region of the genome, typically the mitochondrial cytochrome c oxidase I (COI) gene, to identify species.
Future perspective is missing.
Answer. A final chapter with conclusions and future perspectives was added.
A summary of conclusion is necessary
Answer. A final chapter with conclusions and future perspectives was added.
My suggestion is to revise the manuscript, however, it must have some tables and mechanistic figures, a review study is incomplete without figures.
Answer. We added a figure and a table.
Reviewer 2 Report
Comments and Suggestions for Authors
The manuscript focuses on a very interesting item regarding farmed fish development. It includes wide information and a great number of related manuscripts. However, before a new revision is done, some aspects ought to be performed in order to better present the manuscript.
I would mention the following:
Abstract
The authors ought to mention at first that the manuscript consists of a Review. Otherwise, a possible reader will think that lines 6-18 include very general information and that the current study is only presented in lines 18-19.
Keywords
Include “rearing systems”.
Introduction
As for the Abstract section, this section does present correctly the Review. After reading it, the reader cannot have an opinion of the real content of the manuscript.
Main part
In order to make the manuscript more agreeable, the authors could include some Figures/Tables.
Conclusions
A section like this, or regarding on-coming research, could reinforce the manuscript.
Author Response
The manuscript focuses on a very interesting item regarding farmed fish development. It includes wide information and a great number of related manuscripts. However, before a new revision is done, some aspects ought to be performed in order to better present the manuscript.
I would mention the following:
Abstract
The authors ought to mention at first that the manuscript consists of a Review. Otherwise, a possible reader will think that lines 6-18 include very general information and that the current study is only presented in lines 18-19.
Answer. Thank you for pointing this out. We added a sentence in the beginning of the abstract following your suggestion.
Keywords
Include “rearing systems”.
Answer. We added the keyword
Introduction
As for the Abstract section, this section does present correctly the Review. After reading it, the reader cannot have an opinion of the real content of the manuscript.
Answer. Thank you for your comment. We reconstructed the last paragraph of the introduction on this purpose and we added Figure 1.
Main part
In order to make the manuscript more agreeable, the authors could include some Figures/Tables.
Answer. Thank you for your comment. We added a figure and a table.
Conclusions
A section like this, or regarding on-coming research, could reinforce the manuscript.
Answer. Thank you for your comment. We added a last chapter with Conclusions and future perspectives.
Reviewer 3 Report
Comments and Suggestions for Authors
Minor comments:
Lines 39-42: Explain in detail, how vertical transmission influences the microbiota of marine fish larvae?
Lines 47-51: Explain more in detail about the main bacterial taxa colonizing marine fish larvae during early development?
Lines 75-78: How do live feed organisms contribute to bacterial colonization in fish larvae? Explain brief with recent reference papers
Lines 101-113: What roles do microbial communities play in fish larvae immunity? Any specific for microbial community
Lines 356-378: How do biofilms in aquaculture environments impact microbial diversity and pathogen prevalence?
Lines 336-337: What are the dominant bacterial taxa in different aquaculture systems (flow-through vs. recirculating)?
Lines 304-310: How do environmental factors (temperature, salinity, water exchange) affect microbial dynamics?
Lines 409-459: Briefly, explain the limitations of current microbiome analysis techniques in fish larvae research?
Author Response
Reviewer 3.
Lines 39-42: Explain in detail, how vertical transmission influences the microbiota of marine fish larvae?
Answer. Thank you for your comment. The vertical transmission process was better explained in lines 52-57.
Lines 47-51: Explain more in detail about the main bacterial taxa colonizing marine fish larvae during early development?
Answer. Thank you for your comment. The bacterial taxa in the fish larvae gut microbiota are stated in, Lines 268-270 "Many of the bacteria colonizing the larvae are commensal or beneficial .....Bacterial taxa such as Vibrionales, Actinomycales, Alteromonadales, and Flavobacteriales, include opportunistic...cod larvae [32]", Lines 395-396 "Rotifer diets can....Vibrio members in the guts of post-larval turbot [78], Lines 409-411 "In the early life stages .... predominant in fish guts...[83]. Similar trends have been noted ....Senegalese sole [11]", Line288-289 "Rhodobacteraceae (70.1% relative abundance [7] with some OTUs being shared between larvae and Artemia".
Lines 75-78: How do live feed organisms contribute to bacterial colonization in fish larvae? Explain brief with recent reference papers.
Answer. Thank you for your comment. There is an extensive section (3.2) of the article dealing specifically with this topic.
Lines 101-113: What roles do microbial communities play in fish larvae immunity? Any specific for microbial community.
Answer. Thank you for your comment. Microbial communities play a key role as they stimulate the immune system of the fish in general. This is explained in the section referred to by the reviewer.
Lines 356-378: How do biofilms in aquaculture environments impact microbial diversity and pathogen prevalence?
Answer. Thank you for your comment. There is an exchange of biofilm and water microbiota. As mentioned in the article 377-378 biofilms inhibit the action of disinfectant, so pathogen may prevail a treatment with disinfectant if present in the biofilm.
Lines 336-337: What are the dominant bacterial taxa in different aquaculture systems (flow-through vs. recirculating)?
Answer. Thank you for your comment. There are papers like the ones by Attramadal and Vadstein (papers with reference 28, 100, 102, 103) which focus on this issue and a comparison of flow through versus microbial maturation and recirculation) using cod as a model species. Our paper is more general and not focused on one of these approaches or one species.
Lines 304-310: How do environmental factors (temperature, salinity, water exchange) affect microbial dynamics?
Answer. Thank you for your comment. Sudden changes will affect fish larvae, so these are generally avoided in larval rearing. Environmental factors influence microbial dynamics, but this was not an issue in this article as they vary greatly across species cultured.
Lines 409-459: Briefly, explain the limitations of current microbiome analysis techniques in fish larvae research?
Answer. Thank you for your comment. There is a section of the article on this issue (section 4).
Round 2
Reviewer 1 Report
Comments and Suggestions for Authors
I accept the manuscript in present form. Congratulations